# Increasing Trend of Nontuberculous Mycobacteria Isolation in a Referral Clinical Laboratory in South Korea

**DOI:** 10.3390/medicina57070720

**Published:** 2021-07-16

**Authors:** Yu-Mi Lee, Min-Jin Kim, Young-Jin Kim

**Affiliations:** 1Division of Infectious Diseases, Department of Internal Medicine, Kyung Hee University Hospital, Kyung Hee University School of Medicine, Seoul 02447, Korea; cristal156@hanmail.net; 2Seegene Medical Foundation, Seoul 04805, Korea; lithium2864@mf.seegene.com; 3Department of Laboratory Medicine, Kyung Hee University Hospital, Kyung Hee University School of Medicine, Seoul 02447, Korea

**Keywords:** isolation, species, nontuberculous mycobacteria, tuberculosis, laboratory, Korea

## Abstract

*Background and Objectives*: Nontuberculous mycobacteria (NTM) infections are increasing worldwide. We evaluated the annual trends of NTM isolation on acid-fast bacillus (AFB) culture, compared to that of *Mycobacterium tuberculosis*, and the distribution of NTM species nationwide in South Korea. *Materials and Methods*: The study was conducted in a diagnostic center that is a major referral laboratory for the diagnosis of tuberculosis and NTM in South Korea. All laboratory results of AFB culture from January 2014 to December 2019 were collected. All NTM identified were definitively identified to the species level. *Results*: A total of 345,871 tests were performed for the diagnosis of mycobacteria. The isolation rate of NTM and *M. tuberculosis* was 3.7% (12,969 cases) and 4.4% (15,081 cases), respectively. The annual isolation rate of NTM increased gradually from 2.7% in 2014 to 4.8% in 2019, whereas that of *M. tuberculosis* decreased from 6.2% to 3.3%. There were 4988 cases of NTM identified to the species level. *M. avium* complex (MAC) was the most common species isolated from pulmonary sites (59.8%), followed by *M. gordonae* (9.2%), *M. abscessus* (7.0%), and *M. fortuitum* (5.5%). Extrapulmonary NTM were identified in 29 cases (0.6%). MAC was also the most common NTM species isolated from extrapulmonary sites (65.5%), followed by *M. kansasii* (10.3%), *M. abscessus* (6.9%), and *M. fortuitum* (6.9%). *Conclusions*: The annual isolation rate of NTM has increased gradually, whereas that of *M. tuberculosis* has decreased. Follow-up studies of the increases in NTM identification and NTM infections in South Korea are needed.

## 1. Introduction

Nontuberculous mycobacteria (NTM) are ubiquitous environmental organisms in soil and water [1]. Over 180 NTM species have been identified, and one-third are associated with human infections [2]. Advances in diagnostic techniques of NTM along with the increased prevalence of immunocompromised individuals have resulted in the gradual increase in the incidence of NTM disease worldwide [3]. Pulmonary infections account for 90% of NTM infections. The prevalence of pulmonary NTM disease is approximately 1–15 per 100,000 person-years in the United States [4]. The prevalence of NTM disease per 100,000 people in South Korea increased from 9.4 in 2009 to 36.1 in 2016 [5]. The reported incidence of NTM lung disease in South Korea has risen from 1.2 to 4.38 until 2010 to 4.8 in 2016 [6,7].

NTM are most frequently isolated from pulmonary sites [8]. The isolation rates of pulmonary NTM range from 1.3 to 22.2 per 100,000 persons, varying according to geographic area, race, and underlying conditions [9]. NTM are less commonly isolated from extrapulmonary sites, such as cerebrospinal fluid, joint fluid, soft tissue, muscle, and bone. Many reports have described the distribution of NTM species in pulmonary sites. *Mycobacterium avium complex* (MAC) is recognized as the most common species of NTM lung disease [9,10,11]. However, previous studies were performed in single tertiary-care hospitals. In addition, knowledge of the distribution of NTM species for extrapulmonary infections is limited. The objective of our study is to evaluate the annual trends of the isolation rate of NTM on acid-fast bacillus (AFB) culture, compared to that of *Mycobacterium tuberculosis*, and the distribution of NTM species in hospitals nationwide in South Korea.

## 2. Materials and Methods

### 2.1. Study Settings

This study was conducted in one of the largest referral laboratories of South Korea (Seegene Medical Foundation in Seoul). This lab performs diagnostic tests on samples received from 185 medical institutions through 40 branches nationwide. For the diagnosis of tuberculosis and NTM, two types of samples are required. We collected all results of AFB culture and identified NTM from September 2014 to April 2019 recorded at the referral laboratory. Samples were from five zones in South Korea: Seoul and Gyunggi, Gangwon, Chungcheong, Gyengsang, and Jeolla. The medical records of all subjects were retrospectively reviewed.

### 2.2. Mycobacterial Culture and Identification

The non-sterile specimens were decontaminated by N-acetyl-L-cysteine-2% sodium hydroxide (NALC-NaOH) for 18 min. Sterile samples were not processed for decontamination. The prepared samples were inoculated into a mycobacterial growth indicator tube (MGIT 960 system; Becton Dickinson, Sparks, MD, USA) for AFB culture and on 2% Ogawa agar (Asanpharm, Seoul, Korea). They were incubated for 6 and 8 weeks, respectively. For AFB culture-positive samples, the *M. tuberculosis* complex was first screened using Anyplex™ MTB/NTM Real-time Detection (Seegene Inc., Seoul, Korea) according to the manufacturer’s instructions. If the test result confirmed NTM, the laboratory contacted the particular institution concerning additional identification of NTM species. If there was a request for identification of NTM species, the AFB culture-positive medium was subjected to further identification to the species level using a GenoType Mycobacterium CM/AS assay (Hain Lifescience, Nehren, Germany) according to the manufacturer’s instructions.

### 2.3. Isolation Rate of NTM and Species Distribution

The annual isolation rate of NTM was calculated as the number of NTM cases confirmed by AFB culture divided by the total number of culture tests performed in each year or study period. The results were expressed as percentages.

## 3. Results

### 3.1. Annual Trends of Isolation Rate of NTM and M. tuberculosis

There were 345,871 cultures performed for the diagnosis of mycobacteria. These included 14,759 contamination cases (4.3%). The isolation rate of NTM and *M. tuberculosis* was 3.7% (12,969 cases) and 4.4% (15,081 cases), respectively. The annual isolation rate of NTM increased gradually from 2.7% to 4.8% (*p* < 0.001), whereas that of *M. tuberculosis* decreased from 6.2% to 3.3% (*p* < 0.001) (Figure 1). The isolation rate of NTM (3.9%) was similar to that of *M. tuberculosis* (3.9%) in 2018 and higher (4.8%) than that of *M. tuberculosis* (3.3%) in 2019.

Among the 12,969 NTM cases, 4988 (38.5%) were further analyzed to the species level. The distribution of NTM species (except *M. gordonae*) according to pulmonary and extrapulmonary sites is presented in Figure 2. The NTM isolated from pulmonary sites accounted for 99.5% (4962 cases). Among the NTM identified, MAC (59.8%) was the most common species isolated from pulmonary sites, involving *M. intracellulare* (42.3%) and *M. avium* (17.5%). Rapid growing NTM, such as *M. abscessus*, *M. fortuitum*, and *M. chelonae*, comprised 16% of NTM isolated from pulmonary sites. The distribution trend of NTM species is shown in Figure 3.

Extrapulmonary NTM were identified as 0.5% (26 cases). MAC (61.5%) was also the most common NTM species isolated from extrapulmonary sites, followed by *M. kansasii* (11.5%), *M. abscessus* (7.7%), and *M. fortuitum* (7.7%). Eight cases were isolated from soft tissues. These involved *M. intracellulare* in two cases, and *M. avium*, *M. abscessus*, *M. chelonae*, *M. kansasii*, *M. fortuitum*, and *M. ulcerans* (one case each). Three cases were isolated from urine (*M. avium*, *M. intracellulare*, and *M. fortuitum*). One case of *M. abscessus* was isolated from joint fluid. The extrapulmonary site of isolation was not determined in the remaining 14 cases.

### 3.2. Trends in Distribution of NTM Species from Pulmonary Sites by Year

Trends in the distribution of NTM species from pulmonary sites from 2014 to 2019 are presented in Figure 3. The isolation rate of MAC increased from 2017 (*p* < 0.001). The isolation rate of *M. abscessus* decreased from 2015 (*p* < 0.001), with a similar rate thereafter. The other rapid growing mycobacteria were isolated in comparable proportions by year.

### 3.3. Regional Distribution of NTM Species from Pulmonary Sites

The regional distribution of the top five NTM species isolated from pulmonary sites is presented in Figure 4. MAC was the most common species in all five regions of South Korea. *M. abscessus* was the most common rapid growing mycobacteria in all provinces. *M. fortuitum* accounted for 4 to 10% of NTM species nationwide. *M. gordonae* accounted for 15.2% (*n* = 339) in Gyeongsang Province (data not shown). The isolation frequencies of *M. gordonae* were 2 to 5% in the other provinces. Notably, of the 339 cases of *M. gordonae* found in Gyeongsang Province, 313 were identified in one institution specializing in pneumoconiosis treatment.

## 4. Discussion

The isolation rate of NTM was 3.3% during the study period. The annual isolation rate of NTM increased gradually from 2014 to 2019, whereas that of *M. tuberculosis* decreased. Since 2018, the isolation rate of NTM has exceeded that of *M. tuberculosis*. The increasing NTM isolation rate has also been reported in previous studies in South Korea. One study performed in Seoul chronicled the steadily increasing recovery rate of NTM from respiratory specimens over a 10-year period from 2001 to 2011 [12]. In that study, the recovery rate of NTM exceeded that of *M. tuberculosis* from 2005. In another report from Busan, a continuous increase in the NTM recovery rate between 2009 and 2015 was described. The AFB culture positive rate reached 44.8%, but did not exceed the recovery rate of *M. tuberculosis* [13].

The increase in the numbers of patients with previous lung diseases has influenced the isolation frequency of NTM [14]. The increase in immunocompromised hosts may have implications for the isolation rate of NTM, despite the limited evidence to date [14,15]. In addition, increased awareness of NTM as potential pathogens and advances in diagnostic methods for NTM identification also may have influenced the changing prevalence of NTM isolation. In South Korea, the incidence rate of *M. tuberculosis* is approximately 59 per 100,000 persons, an intermediate-risk burden [16]. Therefore, the positive result of acid-fast bacilli staining has been considered positive for *M. tuberculosis*. However, NTM should be considered as the pathogens before empiric treatment for *M. tuberculosis*, considering the recent increase in the isolation rate of NTM.

MAC was the most common NTM species in our study, irrespective of infection site and geographic region. The distribution of NTM species varies widely across countries and geographic regions [11]. MAC accounted for the majority of the NTM species in the United States, Australia, Japan, Thailand, Singapore, India, and Taiwan. *M. chelonae* and *M. gordonae* are the most prevalent species identified from respiratory specimens in China and Hong Kong, respectively [11]. *M. kansasii* is the most common species in Poland and Slovakia [11]. The proportion of rapid growing mycobacteria was approximately 14 to 20% in previous reports in South Korea, similar to our result [7,10,17]. Rapidly growing mycobacteria are the most frequently isolated species in certain countries, such as the United Kingdom and Greece [11]. *M. abscessus* was the most common rapidly growing mycobacteria isolated from respiratory specimens in previous studies, similar to the present study [18,19]. However, *M. abscessus* was less common in our study than in previous reports, comprising approximately 30% of the NTM isolates. The past and present proportions of NTM species were not markedly different in South Korea. There were also no perceivable differences in regional differences of NTM species between the five districts of South Korea. MAC was the most common pathogen, and *M. abscessus* and *M. fortuitum* were the most frequently isolated rapidly growing mycobacteria in most districts.

Notably, the isolation rate of *M. gordonae* was extraordinarily high (20.2%, 313/1549) in one hospital specializing in the treatment of pneumoconiosis compared to in other institutions. In this hospital, the median (range) age of patients (all male) with *M. gordonae* isolation was high (74 years, range 68–78 years). Sixty-five patients harbored two or more *M. gordonae* that were repeatedly isolated and 28 patients harbored three or more isolates. Considering the characteristics of this hospital and the patients, the possibility that *M. gordonae* is a true pathogen can be considered. However, with our data, there is a limitation in checking whether these patients were actually diagnosed with both pneumoconiosis and *M. gordonae* infection. In addition, as the number of *M. gordonae* isolates in this hospital decreased significantly to four in 2018 and 15 in 2019, the possibility of contamination by *M. gordonae* in the past cannot be excluded.

Our study has other limitations. First, it is difficult to access the clinical data of the study patients. Caution is needed in presuming that the isolation rate of NTM does not represent the rate of clinical infections. In addition, some redundant test results may have affected the analysis of the isolation rate of NTM because the denominator of the isolation rate of NTM was the total number of culture tests performed during the study period. In this regard, the results should be interpreted with caution. Second, some of the NTM species may be missed by the GenoType Mycobacterium CM/AS assay for the identification of NTM species. Third, a small number of extrapulmonary specimens were included in our study. Fourth, it was unfeasible to conduct the comparison of the isolation prevalence across the regions because the total number of culture tests performed according to the regions was unavailable. Finally, the species distribution data may differ slightly from reality as not all NTM culture cases were requested for NTM identification. Despite these limitations, the data may be helpful in understanding the isolation rate of NTM and the trend of distribution of NTM species chronologically and nationally.

## 5. Conclusions

The isolation rate of NTM in one referral laboratory has gradually increased. Since 2018, this rate has exceeded that of *M. tuberculosis*. Further studies are needed on the trend of increased NTM isolation and NTM infections across South Korea.

## Figures and Tables

**Figure 1 medicina-57-00720-f001:**
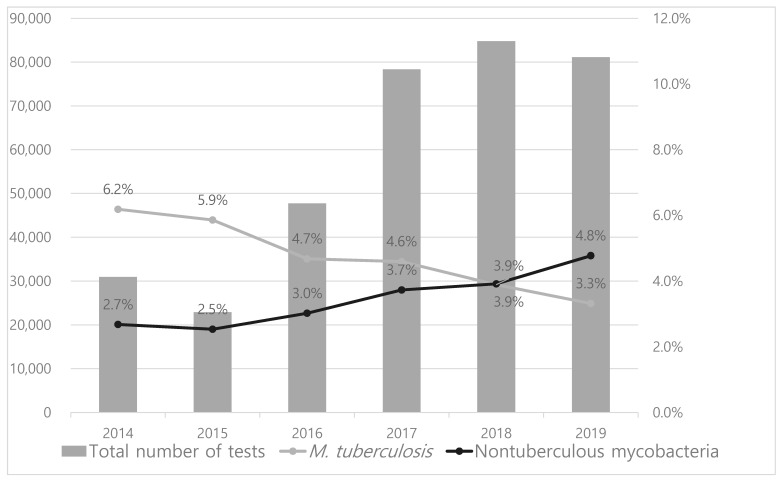
Annual trends of isolation rate of nontuberculous mycobacteria, compared to that of *M. tuberculosis*. *M. tuberculosis*: *Mycobacterium tuberculosis*.

**Figure 2 medicina-57-00720-f002:**
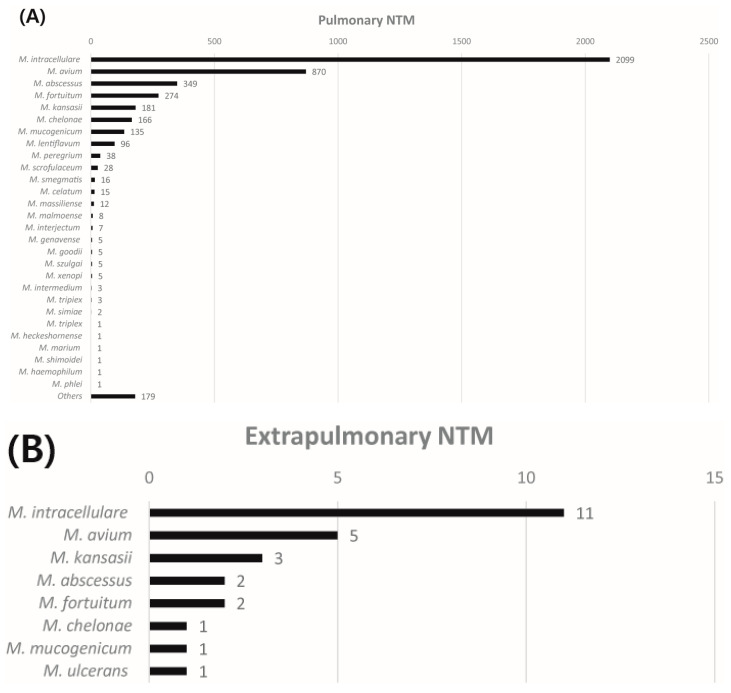
Distribution of nontuberculous mycobacteria species according to (**A**) pulmonary sites and (**B**) extrapulmonary sites. NTM: Nontuberculous mycobacteria.

**Figure 3 medicina-57-00720-f003:**
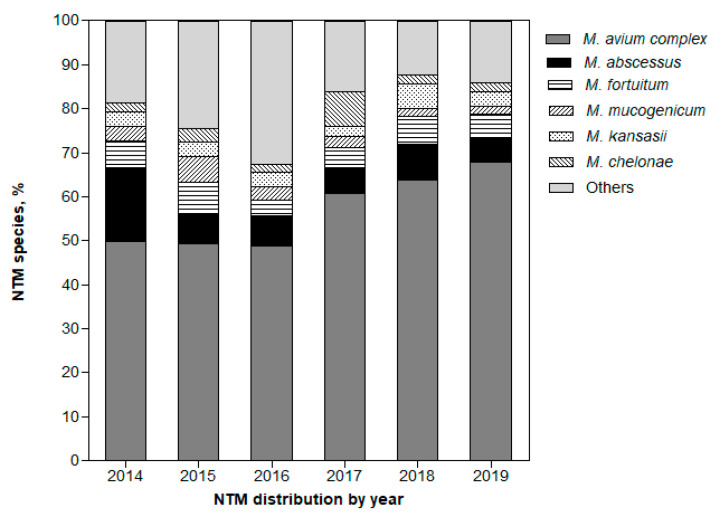
Trends in distribution of nontuberculous mycobacteria species isolated from pulmonary sites from 2014 to 2019.

**Figure 4 medicina-57-00720-f004:**
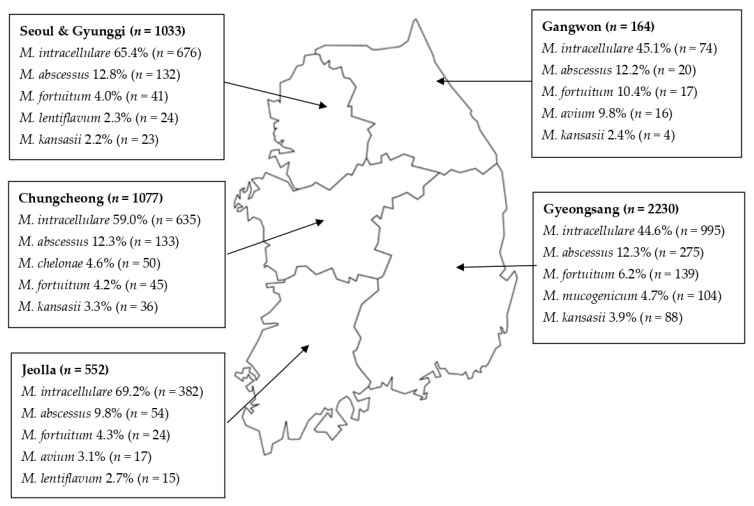
Regional distribution of nontuberculous mycobacteria species isolated from pulmonary sites in South Korea.

## Data Availability

The datasets used and/or analyzed during the current study are available from the corresponding author on reasonable request.

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
