# Peer review of "Increasing Trend of Nontuberculous Mycobacteria Isolation in a Referral Clinical Laboratory in South Korea"

_medicina, 2021, doi:10.3390/medicina57070720_

Round 1
Reviewer 1 Report
This study provides an important overview of the trends regarding M. TB and NTM over a recent period. These data are timely and relevant to understanding the current burden of NTM in a country with a historically moderate burden of TB
- Some citations are updated. The most up to date citations should be used
- Where does the prevalence of 10- 40 per 100,000 come from? This should be updated and a the appropriate measure and time scale should be mentioned. Prevalence? Per year? Per period?
- M gordonae should be left out of the figures and tables, and that species is typically considered non pathogenic
- Did the isolation prevalence overall vary across regions?
- Regarding the statement in the discussion that the increasing isolation rate is a result of increasing numbers of immucompromised persons- that has not been clearly shown. I suggest modifying the statement to say that it could be a reason, but the evidence is minimal.
- The authors should clarify that their estimates represent isolates and not people, and therefore if the same person is getting tested repeatedly for NTM, that could result in an artifactual increase in isolation. That reason is unlikely the cause of the increase in the isolation rate, but should be mentioned as a limitation, that the denominator is tests and not people . The authors do mention that caution should be used in interpreting these rates, not to presume that these are rates of clinical infections, but another sentence would be helpful.
Author Response
Date: 14 JUL 2021
Re: MS medicina-1278999, "Increasing Trend of Nontuberculous Mycobacteria Isolation in a Referral Clinical Laboratory in South Korea”
On behalf of the authors, I would like to thank you and the reviewers for the valuable and helpful comments on our submitted manuscript, “Increasing Trend of Nontuberculous Mycobacteria Isolation in a Referral Clinical Laboratory in South Korea”. We have carefully reviewed the comments and have revised the manuscript accordingly. Our responses are given in a point-by-point manner as shown below, and changes to the manuscript are shown in highlight.
Reviewer #1’s comments:1.
Some citations are updated. The most up to date citations should be used.Where does the prevalence of 10- 40 per 100,000 come from? This should be updated and a the appropriate measure and time scale should be mentioned. Prevalence? Per year? Per period?
I would like to thank you for the valuable and helpful comments on our submitted manuscript. There were some differences in the prevalence of pulmonary NTM disease in North America according to the investigation period and regions. Iseman et al. reported that the prevalence of pulmonary NTM disease in US was in the range of 14 to 35 per 100,000 persons (AJRCCM 2008;178:999-1001). Kendall et al. reported the epidemiology of pulmonary NTM infections. In this study, the rates of pulmonary NTM diseases range from 1 to 15 per 100,000 person-years (Semin Respir Crit Care Med 2013;34:87–94). In other study, the prevalence increased from 20 to 47/100,000 persons from 1997 to 2007 (AJRCCM 2012;185(8):881-886). As the reviewer’s comment, we have revised this sentence clearly in the Introduction section as follows.
Introduction section:
From : “The prevalence is approximately 10 to 40 cases per 100,000 in the United States.”
To: “The prevalence of pulmonary NTM disease is approximately 1-15 per 100,000 person-years in the United States [Semin Respir Crit Care Med 2013;34:87–94].”
2. M gordonae should be left out of the figures and tables, and that species is typically considered non pathogenic
As the reviewer’s comment, we have revised the Figure 2, 3, and 4 as follows.
Figure 2. Distribution of Nontuberculous mycobacteria species according to (A) pulmonary sites and (B) extrapulmonary sites.
We have revised the sentence in the Results section as follows.“The distribution of NTM species (except M. gordonae) according to pulmonary and ex-trapulmonary sites is presented in Figure 2.”
Figure 3. Trends in distribution of Nontuberculous mycobacteria species isolated from pulmonary sites from 2014 to 2019.
Figure 4. Regional distribution of Nontuberculous mycobacteria species isolated from pulmonary sites in South Korea.
We have revised the sentence in the Results section as follows.
From: “The isolation rate of MAC increased from 2017 (P<0.001), whereas that of M. gordonae decreased from 2017 (P<0.001).”
To: “The isolation rate of MAC increased from 2017 (P<0.001).”
3. Did the isolation prevalence overall vary across regions?
Unfortunately, we could only access the data for distribution of the identified species according to the regions. It was unfeasible to conduct the comparison of the isolation prevalence across the regions, because the total number of culture tests performed according to the regions was unavailable. We added this information as a limitation in the Discussion section as follows.
“Fourth, it was unfeasible to conduct the comparison of the isolation prevalence across the regions, because the total number of culture tests performed according to the regions was unavailable.”
4. Regarding the statement in the discussion that the increasing isolation rate is a result of increasing numbers of immucompromised persons- that has not been clearly shown. I suggest modifying the statement to say that it could be a reason, but the evidence is minimal.
As the reviewer’s comment, we revised this sentence in the Discussion section as follows.
From: “The increase in the numbers of immunocompromised hosts and patients with previous lung diseases has influenced the isolation frequency of NTM.”
To: “The increase in the numbers of patients with previous lung diseases has influenced the isolation frequency of NTM. The increase of the immunocompromised hosts may have an implication on the isolation rate of NTM, despite of the limited evidence to date [J Infect 2007;55(6):484-7, Respirology (Carlton, Vic) 2009;14(1):12-26].”
5. The authors should clarify that their estimates represent isolates and not people, and therefore if the same person is getting tested repeatedly for NTM, that could result in an artifactual increase in isolation. That reason is unlikely the cause of the increase in the isolation rate, but should be mentioned as a limitation, that the denominator is tests and not people . The authors do mention that caution should be used in interpreting these rates, not to presume that these are rates of clinical infections, but another sentence would be helpful.
As the reviewer’s comment, we added this information as a limitation in the Discussion section as follows.
“In addition, some redundant test results may have affected the analysis of the isolation rate of NTM, because the denominator of the isolation rate of NTM was the total number of culture tests performed during the study period. In this regard, the results should be interpreted with caution.”
We believe we have addressed all questions and comments in a suitable fashion, but would be happy to provide further information or revision if necessary.
Sincerely yours,
Young-Jin Kim, M.D.
Department of Laboratory Medicine,
Kyung Hee University School of Medicine
23, Kyungheedae-ro, Dongdaemun-gu,
Seoul, 02447, Republic of Korea
Tel: 82-2-958-8674, Fax: 82-2-968-8609
E-mail: khmclab@gmail.com

Reviewer 2 Report
The article is about the increased detection of NTM in pulmonary and extrapulmonary samples. The article is well written and the study is appropriately designed. The English is adequate. The methods are well elucidated and the results are exhaustive.The discussion is adequate. Congratulations on the paper.
Author Response
Date: 14 JUL 2021
Re: MS medicina-1278999, "Increasing Trend of Nontuberculous Mycobacteria Isolation in a Referral Clinical Laboratory in South Korea”
On behalf of the authors, I would like to thank you and the reviewers for the valuable comments on our submitted manuscript, “Increasing Trend of Nontuberculous Mycobacteria Isolation in a Referral Clinical Laboratory in South Korea”.
Sincerely yours,
Young-Jin Kim, M.D.
Department of Laboratory Medicine,
Kyung Hee University School of Medicine
23, Kyungheedae-ro, Dongdaemun-gu,
Seoul, 02447, Republic of Korea
Tel: 82-2-958-8674, Fax: 82-2-968-8609
E-mail: khmclab@gmail.com